# Hydrodynamics-Informed Neural Network for Simulating Dense Crowd Motion Patterns

## ABSTRACT

With global occurrences of crowd crushes and stampedes, dense crowd simulation has been drawing great attention. In this research, our goal is to simulate dense crowd motions under six classic motion patterns, more specifically, to generate subsequent motions of dense crowds from the given initial states. Since dense crowds share similarities with fluids, such as continuity and fluidity, one common approach for dense crowd simulation is to construct hydrodynamics-based models, which consider dense crowds as fluids, guide crowd motions with Navier-Stokes equations, and conduct dense crowd simulation by solving governing equations. Despite the proposal of these models, dense crowd simulation faces multiple challenges, including the difficulty of directly solving Navier-Stokes equations due to their nonlinear nature, the ignorance of distinctive crowd characteristics which fluids lack, and the gaps in the evaluation and validation of crowd simulation models. To address the above challenges, we build a hydrodynamic model, which captures the crowd physical properties (continuity, fluidity, etc.) with Navier-Stokes equations and reflects the crowd social properties (sociality, personality, etc.) with operators that describe crowd interactions and crowd-environment interactions. To tackle the computational problem, we propose to solve the governing equation based on Navier-Stokes equations using neural networks, and introduce the Hydrodynamics-Informed Neural Network (HINN) which preserves the structure of the governing equation in its network architecture. To facilitate the evaluation, we construct a new dense crowd motion video dataset called Dense Crowd Flow Dataset (DCFD), containing six classic motion patterns (line, curve, circle, cross, cluster and scatter) and 457 video clips, which can serve as the groundtruths for various objective metrics. Numerous experiments are conducted using HINN to simulate dense crowd motions under six motion patterns with video clips from DCFD. Objective evaluation metrics that concerns authenticity, fidelity and diversity demonstrate the superior performance of our model in dense crowd simulation compared to other simulation models.

## CCS CONCEPTS

• **Computing methodologies** → **Computer vision**.

## KEYWORDS

Crowd Simulation, Dense Crowd Motion, Hydrodynamic Model

**ACM Reference Format:**
Anonymous Author(s). 2024. Hydrodynamics-Informed Neural Network for Simulating Dense Crowd Motion Patterns. In *Proceedings of the 32nd ACM International Conference on Multimedia (MM'24), October 28-November 1, 2024, Melbourne, Australia.* ACM, New York, NY, USA, 9 pages. https://doi.org/XXXXXXX.XXXXXXX

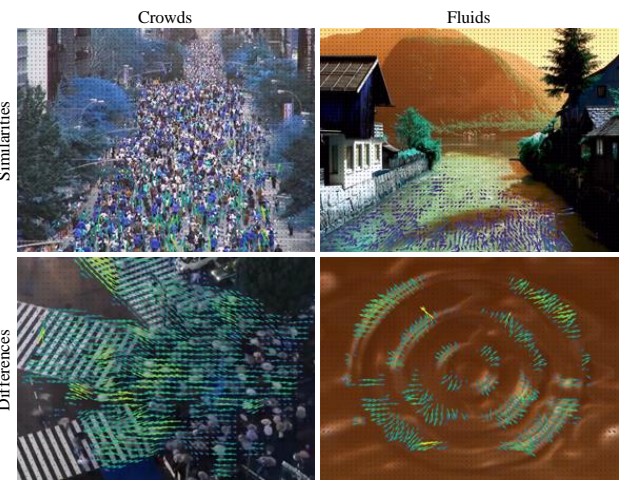

Figure 1: Crowds and fluids exhibit similarities as well as differences. The first row highlights that both crowds and fluids share continuity and fluidity (physical properties), while the second row suggests that crowds can respond to the environment and possess sociality and personality (social properties) which fluids lack.

## 1 INTRODUCTION

Crowd simulation has long been popular within computer vision for its extensive applications in fields of urban planning [10, 30], emergency evacuation [3, 14], virtual reality [32, 45], video games [48, 53], and more. With crowd crushes and stampedes occurring frequently around the world, there is an increasingly urgent desire for dense crowd simulation in particular. Through dense crowd simulation, it becomes possible to predict crowd motions, prevent potential accidents, and protect public safety.

Models for crowd simulation can be generally categorized into microscopic (rule-based or agent-based) models and macroscopic (hydrodynamics-based) models. Classic microscopic models, such as Boids [41], Vicsek [51] and Social Force Model (SFM) [15], require each crowd member or agent to perceive and react individually to the environment and others based on certain rules. Along this line of thought, follow-up researches extend these models by considering aspects of evacuation panic [5, 14, 34], collision avoidance [35, 49], human psychology [8, 9, 37], etc. These models are normally applied

for small-scale crowd simulation and are not suitable for large-scale due to increased computational complexity and challenges in capturing individual behaviors.

On the contrary, macroscopic models consider crowds as fluid-like entities, utilizing hydrodynamics to handle the continuity and fluidity of crowd motions, making them more suitable for large-scale or dense crowd simulation. Proposed by Henderson [16] and supplemented by Bradley [4], this approach to constructing hydrodynamic models for dense crowds involves regarding dense crowds as fluids, guiding crowd motions with Navier-Stokes equations, and conducting dense crowd simulation by solving the governing equations.

Despite the advancement of hydrodynamic models, we are facing three challenges for dense crowd simulation. Firstly, due to the nonlinear nature of Navier-Stokes equations, the solving process remains complicated whether for crowds or fluids. Previous works [43, 50, 55] approximate the solution through Smoothed Particle Hydrodynamics (SPH) [12], which is a computational method that considers crowds or fluids as assemblies of individuals or elements with associated properties and uses smoothing kernels to approach their interactions. However, SPH compromises simulation accuracy and introduces computational complexity as it is particle-based. Secondly, as shown in Fig.1, other than similarities shared with fluids (physical properties), crowds possess characteristics (social properties) such as sociality and personality, which makes it difficult to accomplish the simulation with pure hydrodynamics. Thirdly, there are gaps in the evaluation of crowd simulation, particularly for large-scale or dense crowds. Previous researches normally rely on metrics such as time consumption [7, 33], computational complexity [39, 47], and subjective judgment [11, 55]. However, these metrics are not convincing enough and fail to reflect authenticity, fidelity and diversity of the simulation.

To address the above challenges, we construct a hydrodynamic model concerning both the physical and social properties of crowds. We guide crowd motions with the momentum equation of Navier-Stokes equations to capture physical properties, and incorporate additional forces consistent with crowd interactions and crowd-environment interactions into the governing equation to reflect social properties. Then, inspired by Partial Differential Equation (PDE)-Preserved Neural Network (PPNN) [27], we propose to solve the governing equation using neural networks and introduce the Hydrodynamics-Informed Neural Network (HINN) which preserves the structure of the governing equation in its network architecture. To facilitate the evaluation, we collect hundreds of video clips of six classic dense crowd motion patterns, including line, curve, circle, cross, cluster and scatter, to construct a real-world dense crowd motion video dataset called the Dense Crowd Flow Dataset (DCFD). The construction of DCFD enables us to employ objective metrics about authenticity, fidelity and diversity for evaluation, including Inception Score (IS) [42], Fréchet Inception Distance (FID) [17] and Structural Similarity (SSIM) [52], by serving as the groundtruths. We conduct numerous experiments using HINN to simulate dense crowd motions under the six classic motion patterns with video clips from DCFD, and evaluate the performances of our model and other simulation models with objective metrics. Our contributions are summarized as follows:

- We propose a hydrodynamic model which considers both the similarities shared between crowds and fluids, as well as the unique characteristics possessed by crowds, namely physical and social properties.
- We design three operators to describe interactions within dense crowds and between crowds and environments. These operators can be applied to simulate six classic dense crowd motion patterns.
- We construct a new real-world dense crowd video dataset which contains six classic dense crowd motion patterns. Experiments demonstrate the effectiveness of our approach compared with other simulation models by using objective evaluation metrics.

## 2 RELATED WORK

### 2.1 Hydrodynamics-based Crowd Simulation

As crowds and fluids share continuity and fluidity, hydrodynamics-based models aim to reflect crowd physical properties by drawing upon insights from fluid dynamics. This approach was proposed by Henderson [16], regarding loose crowds as gas, dense crowds as liquid, and loose-dense transitions as phase transformation. To go one step further, Bradley [4] guided dense crowd motions with Navier-Stokes equations and accomplished dense crowd simulation by solving the governing equations. On this basis, Hughes [20] suggested to discard terms with little effect and introduce additional terms to the governing equations based on the specific situation. For the difficulty of solving Navier-Stokes equations directly, follow-up researchers, such as Yuan [55] and van Toll [50], mostly used SPH [12] to approximate the solution. However, SPH on the one hand compromises accuracy, and on the other hand introduces computational complexity as it is particle-based. Hence, the main problem of such models lies in the absence of an accurate yet simple computational method for Navier-Stokes equations.

In this research, we construct a hydrodynamic model to reflect physical properties of dense crowd motions under the guidance of Navier-Stokes equations. As a complement, we introduce additional forces into the governing equation to capture social properties of dense crowds. To tackle the aforementioned computational problem, we propose to solve the governing equation with PDE-solving neural networks.

### 2.2 PDE-Solving Neural Networks

Methods of solving PDE with neural networks can be basically divided as data-driven and physics-informed. Data-driven neural networks, including Convolutional Neural Networks (CNN) [13, 23] and PDE-Net [28], learn differential operators using convolutional kernels and fit the nonlinear response function by neural networks. Such networks rely heavily on large amounts of data, struggle with error accumulation and have poor generalizability. To tackle these problems, Physics-Informed Neural Networks (PINNs) [40] and its variants [21, 31] employ Deep Neural Networks (DNN) for their well-known capability as universal function approximators [18], exploit automatic differentiation, and utilize the prior physics knowledge by incorporating PDE residuals into the loss function. Besides, physics-informed variants of neural operators, such as DeepONet [29] and Fourier Neural Operator (FNO) [26], learn the

**Figure 2: The schematic diagram of the proposed Hydrodynamics-Informed Neural Network (HINN), which consists of a trainable network and a plug-in module that preserves the structure of the governing equation. The trainable network adopts a Convolutional Residual Network (ConvResNet) [27] due to its demonstrated effectiveness in previous work on PPNN. The plug-in module, called Hydrodynamics-Informed Module (HIM), uses the bilinear down-sampling and the bicubic up-sampling to establish a multi-resolution setting. The governing equation for dense crowd simulation are calculated by two physical operators and three social operators, which reflects the physical and social properties at the same time.**

mapping from functional parametric dependence to the solution, thus learn an entire family of PDE, instead of solving one instance. However, this method of embedding PDE residuals into the loss function depends largely on extensive adjustments of the hyper-parameters to weigh each loss term. To this end, PDE-Preserved Neural Network (PPNN) [27] embeds PDE into the neural network architecture via connections between PDE operators and network structures, which improves the generalizability as well as long-term prediction accuracy at the same time.

In this research, to address the aforementioned computational challenge faced by hydrodynamics-based models, inspired by PPNN, we propose HINN to solve our governing equation. HINN preserves the structure of our governing equation based on Navier-Stokes equations for dense crowd simulation within a plug-in module called Hydrodynamics-Informed Module (HIM).

## 3 METHODOLOGY

### 3.1 Overview

Our goal is to build a hydrodynamic model that considers both the physical and social properties for dense crowd simulation, more specifically, for generating subsequent motions of dense crowds from the given initial states. For a dense crowd video clip with $T+1$ frames, $F = [f_0, f_1, \ldots, f_T]$, dense optical flows can be computed between consecutive frames. These optical flows approximately provide the velocity field $\mathbf{U} = [\mathbf{u}_0, \mathbf{u}_1, \ldots, \mathbf{u}_{T-1}]$, where $\mathbf{u}_t$ is the velocity between two consecutive frames $f_t$ and $f_{t+1}$. Taking the initial velocity $\mathbf{u}_0$ as input, the hydrodynamic model can predict the velocity $\hat{\mathbf{u}}_1$ for the next time-step. Subsequently, for any time-step $t > 0$, the hydrodynamic model can predict the succeeding velocity $\hat{\mathbf{u}}_{t+1}$ based on the previously predicted velocity $\hat{\mathbf{u}}_t$. Consequently, the predicted velocity field of this video clip can be denoted as $\hat{\mathbf{U}} = [\hat{\mathbf{u}}_0, \hat{\mathbf{u}}_1, \ldots, \hat{\mathbf{u}}_{T-1}]$, where $\hat{\mathbf{u}}_0 = \mathbf{u}_0$. Then, our target is to minimize the distance between groudtruths $\mathbf{U}$ and predictions $\hat{\mathbf{U}}$,

which can be reformulated as an optimization problem:

$$\min_{\hat{\mathbf{U}}=[\hat{\mathbf{u}}_0, \hat{\mathbf{u}}_1, \ldots, \hat{\mathbf{u}}_{T-1}]} Distance\left(\mathbf{U}, \hat{\mathbf{U}}\right) \quad (1)$$

where $Distance\left(\cdot, \cdot\right)$ can be various evaluation metrics that measure the distance between groudtruths $\mathbf{U}$ and predictions $\hat{\mathbf{U}}$, including Huber loss [19], IS, FID, SSIM, etc.

### 3.2 Framework

To simulate dense crowd motions, we construct a hydrodynamic model that guides the crowd motions with a hydrodynamics-based governing equation and solves the governing equation using a next-step prediction network called HINN. As shown in Fig.2, HINN contains two parts: a trainable network that can be any predictive network [13, 23, 46], but we adopt ConvResNet here because of its effectiveness demonstrated in PPNN; and a plug-in module called HIM, which preserves the structure of the governing equation. At any time-step $t$, taking the previously predicted velocity $\hat{\mathbf{u}}_t$ as the input, HIM converts it to low resolution by using bilinear down-sampling, extracts the hidden feature preserved by the governing equation in the low-resolution space, and uses bicubic up-sampling to transform extracted feature back into original high-resolution space as $\hat{\mathbf{u}}'_t$. Then, the acceleration-related feature $\hat{\mathbf{u}}'_t$ is attached to the input velocity $\hat{\mathbf{u}}_t$, serving as the combined input for the trainable network to predict the velocity $\hat{\mathbf{u}}_{t+1}$ for the next time-step. This predicted velocity $\hat{\mathbf{u}}_{t+1}$ can further serve as the input again to predict subsequent velocity. After $T$ steps, we obtain $\hat{\mathbf{U}} = [\hat{\mathbf{u}}_0, \hat{\mathbf{u}}_1, \ldots, \hat{\mathbf{u}}_{T-1}]$ that denotes the dense crowd motion generated from the initial states.

For the governing equation, we consider both physical and social properties of crowds. Since crowds share continuity and fluidity with fluids, we capture the crowd physical properties according to the following momentum equation of Navier-Stokes equations for

incompressible fluids:

$$\frac{\partial \mathbf{u}}{\partial t} + (\mathbf{u} \cdot \nabla)\mathbf{u} = \nu \nabla^2 \mathbf{u} - \frac{1}{\rho}\nabla p + \mathbf{f} \tag{2}$$

where $\mathbf{u}$ is the velocity, $\nu$ is the viscosity, $\rho$ is the density, $p$ is the pressure, and $\mathbf{f}$ is the external force. Generally, this equation can be decomposed into five key components: the acceleration $\frac{\partial \mathbf{u}}{\partial t}$, the convective acceleration $(\mathbf{u} \cdot \nabla)\mathbf{u}$, the viscosity force $\nu \nabla^2 \mathbf{u}$, the pressure force $-\frac{1}{\rho}\nabla p$ and the external forces $\mathbf{f}$.

In this research, we regard dense crowd simulation as a two-dimensional problem, in which case the momentum equation of Navier-Stokes equations in horizontal and vertical directions can be expressed as follows:

$$\frac{\partial u}{\partial t} + \left(u\frac{\partial u}{\partial x} + v\frac{\partial u}{\partial y}\right) = \nu\left(\frac{\partial^2 u}{\partial x^2} + \frac{\partial^2 u}{\partial y^2}\right) - \frac{1}{\rho}\frac{\partial p}{\partial x} + f_x \tag{3}$$

$$\frac{\partial v}{\partial t} + \left(u\frac{\partial v}{\partial x} + v\frac{\partial v}{\partial y}\right) = \nu\left(\frac{\partial^2 v}{\partial x^2} + \frac{\partial^2 v}{\partial y^2}\right) - \frac{1}{\rho}\frac{\partial p}{\partial y} + f_y \tag{4}$$

where $u$ and $v$ are the horizontal and vertical components of velocity, $f_x$ and $f_y$ are the horizontal and vertical components of external forces respectively.

Meanwhile, to reflect social properties of dense crowd motions, we introduce additional forces into the equation concerning the characteristics that crowds possess while fluids lack. Combining these two aspects, we formulate our governing equation for dense crowd simulation considering the following terms, where the first two terms reflect the physical properties and the others reflect the social properties.

*Convection force.* Similar to fluids, this term is caused by the uneven distribution of the velocity field. The relative velocity of crowds can lead to collisions or tendencies to avoid collisions, bringing about changes in the velocity field. Thus, we denote the convective force for dense crowds as:

$$f_{con_x} = u\frac{\partial u}{\partial x} + v\frac{\partial u}{\partial y} \quad f_{con_y} = u\frac{\partial v}{\partial x} + v\frac{\partial v}{\partial y} \tag{5}$$

To implement the convective force in HIM, we approximate the first-order differentiation with the Sobel operators:

$$\frac{\partial}{\partial x} = \begin{bmatrix} -1 & 0 & 1 \\ -2 & 0 & 2 \\ -1 & 0 & 1 \end{bmatrix} \quad \frac{\partial}{\partial y} = \begin{bmatrix} -1 & -2 & -1 \\ 0 & 0 & 0 \\ 1 & 2 & 1 \end{bmatrix} \tag{6}$$

*Viscosity force.* Originated from the uneven distribution of velocity field as well, we calculate the viscosity force for dense crowds by:

$$f_{vis_x} = \frac{\partial^2 u}{\partial x^2} + \frac{\partial^2 u}{\partial y^2} \quad f_{vis_y} = \frac{\partial^2 v}{\partial x^2} + \frac{\partial^2 v}{\partial y^2} \tag{7}$$

To implement the viscosity force in HIM, we approach the second-order differentiation using the Laplacian operator:

$$\frac{\partial^2}{\partial x^2} + \frac{\partial^2}{\partial y^2} = \nabla^2 = \begin{bmatrix} 0 & 1 & 0 \\ 1 & -4 & 1 \\ 0 & 1 & 0 \end{bmatrix} \tag{8}$$

*Alignment force.* As mentioned in Boids, pressure from surrounding members can drive individual to move along with them, namely to match their velocity. In this way, we represent the alignment force for dense crowds as the difference to average velocity of the surrounding area:

$$f_{ali_x} = \frac{1}{N}\sum_{i\in\mathring{U}} u_i - u \quad f_{ali_y} = \frac{1}{N}\sum_{i\in\mathring{U}} v_i - v \tag{9}$$

where $\mathring{U}$ is the surrounding area and $N$ is the number of grids contained in the area. The alignment force can be implemented in HIM by using the zero-centered mean filter.

*Navigation force.* Referring to SFM, individuals have their desired velocities toward the destination, which manifests as the tendency for people to follow the steps of those ahead of them in dense crowds. To describe this tendency, the navigation force for dense crowds can be expressed as:

$$f_{nav_x} = \frac{u_e - u}{\tau} \quad f_{nav_y} = \frac{v_e - v}{\tau} \tag{10}$$

where $u_e$ and $v_e$ are the horizontal and vertical components of the desired velocity, which is the velocity of the person ahead, and $\tau$ is the relaxation time. To calculate the navigation force, we design a forward discriminator in HIM that can determine the forward direction of each individual, identify the person ahead, and thus obtain the desired velocity for calculation.

*Cohesion force.* Individuals located at crowd boundaries normally do not break away from the crowds easily, trying to resist dispersion and maintain the overall shape of crowds. To capture this property, we compute the cohesion force for dense crowds by:

$$f_{coh_x} = \begin{cases} 1 & left\ border \\ 0 & inner\ area \\ -1 & right\ border \end{cases} \quad f_{coh_y} = \begin{cases} 1 & upper\ border \\ 0 & inner\ area \\ -1 & lower\ border \end{cases} \tag{11}$$

To compute the cohesion force, in HIM, we employ the Canny edge detector [6] to recognize the crowd boundaries.

To sum up, we propose our governing equation for dense crowd simulation as:

$$f_{acc_x} = \lambda_1 f_{con_x} + \lambda_2 f_{vis_x} + \lambda_3 f_{ali_x} + \lambda_4 f_{nav_x} + \lambda_5 f_{coh_x} \tag{12}$$

$$f_{acc_y} = \lambda_1 f_{con_y} + \lambda_2 f_{vis_y} + \lambda_3 f_{ali_y} + \lambda_4 f_{nav_y} + \lambda_5 f_{coh_y} \tag{13}$$

where $\lambda_1$, $\lambda_2$, $\lambda_3$, $\lambda_4$ and $\lambda_5$ are parameters.

## 3.3 Evaluation Metrics

The evaluation in crowd simulation faces considerable challenges. Previous works normally adopt metrics such as time consumption, computational complexity and subjective judgment, which fail to reflect the authenticity, fidelity and diversity of the simulation. The primary reason behind this issue is the lack of datasets that can act as groundtruths. To this end, we construct a new dense crowd motion video dataset called DCFD containing six different motion patterns. Based on DCFD, we propose to employ the following objective metrics to evaluate the performances of our hydrodynamic model and other simulation models.

*Inception Score.* Originally proposed for Generative Adversarial Networks (GAN), IS evaluates the distinctiveness and variety of generated images based on a pretrained classifier called InceptionV3. To employ IS in dense crowd simulation, we modify the input and output layers of InceptionV3 and retrain the classifier on DCFD with a ratio of eight to two between the training set and the test set. Then, IS for dense crowd simulation can be denoted as:

$$KL(P, Q) = \mathbb{E}_{\hat{\mathbf{U}} \sim P} \left[ \log \left( P(\hat{\mathbf{U}}) \right) - \log \left( Q(\hat{\mathbf{U}}) \right) \right] \quad (14)$$

$$IS(\hat{G}) = \exp \left( \mathbb{E}_{\hat{\mathbf{U}} \sim P_g} KL(p(\hat{y} \mid \hat{\mathbf{U}}), p(\hat{y})) \right) \quad (15)$$

where $KL(\cdot, \cdot)$ is the Kullback-Leibler Divergence (KLD) [24], $\hat{G}$ is the prediction set, and $\hat{y}$ is the motion pattern of each velocity field $\hat{\mathbf{U}}$ given by the classifier.

*Fréchet Inception Distance.* Based on InceptionV3 as well, FID takes the groundtruths into account. FID treats the groundtruth set and prediction set as two Gaussian distributions, calculates their mean and covariance respectively, and measures the distance between the groundtruths and predictions by:

$$FID(G, \hat{G}) = \|m - \hat{m}\|_2^2 + \text{Trace} \left( C + \hat{C} - 2\sqrt{C\hat{C}} \right) \quad (16)$$

where $G$ is the groundtruth set and $\hat{G}$ is the prediction set, $(m, C)$ and $(\hat{m}, \hat{C})$ are their mean and covariance respectively.

*Structural Similarity.* Designed for image quality assessment, SSIM compares groundtruths and predictions in terms of three aspects including the luminance, contrast and structure. To apply SSIM to dense crowd simulation, we represent the velocities using heat maps and compute SSIM by:

$$l(h, \hat{h}) = \frac{2\mu_h \mu_{\hat{h}}}{\mu_h^2 + \mu_{\hat{h}}^2} \quad c(h, \hat{h}) = \frac{2\sigma_h \sigma_{\hat{h}}}{\sigma_h^2 + \sigma_{\hat{h}}^2} \quad s(h, \hat{h}) = \frac{\sigma_{h\hat{h}}}{\sigma_h \sigma_{\hat{h}}} \quad (17)$$

$$SSIM(G, \hat{G}) = \frac{1}{NM} \sum_{i=1}^{N} \sum_{j=0}^{M-1} l(h_{ij}, \hat{h}_{ij}) c(h_{ij}, \hat{h}_{ij}) s(h_{ij}, \hat{h}_{ij}) \quad (18)$$

where $l(\cdot, \cdot)$ is the luminance comparison, $c(\cdot, \cdot)$ is the contrast comparison, and $s(\cdot, \cdot)$ is the structure comparison; $h_{ij}$ and $\hat{h}_{ij}$ are the heat maps corresponding to the velocity between the $j_{th}$ and $(j + 1)_{th}$ frames of the $i_{th}$ video in the groundtruth set and prediction set respectively.

## 4 EXPERIMENTS

### 4.1 Crowd Dataset

To facilitate dense crowd researches as well as the evaluation and validation of crowd simulation models, we construct a new dense crowd motion video dataset called DCFD. DCFD contains dense crowd motions of six classic patterns, including line, curve, circle, cross, cluster, and scatter. These crowd motions are sourced from various scenes, such as marathons, parades, intersections, etc. In total, DCFD consists of 457 video clips, which are all collected from Getty Images[1]. More than ninety percent of these video clips are unique to DCFD and have not been included in previous datasets. Each video clip is cropped into MP4 format with a size of $360 \times 480$ pixels, spanning approximately 120 frames. To utilize these dense

crowd videos in our research, we also calculate the velocity field of each video clip using the optical flows [1, 2]. As is shown in Tab.1, most existing crowd video datasets are either not dense enough or contain very few videos. Our DCFD surpasses these datasets considering the crowd density, number of video clips and crowd motion patterns, and notably contains velocity fields for all video clips. All the video clips and velocity fields are sorted and labeled manually by multiple annotators according to crowd motion patterns and are available in the dataset.

**Table 1: Comparison between existing crowd datasets and our DCFD in terms of the crowd density, number of video clips, number of motion patterns and velocity field. Note that we consider consecutive frames as one video when counting the number of video clips.**

| Dataset | Density | Videos | Patterns | Velocity |
|---|---|---|---|---|
| GC [54] | loose | 1 | × | × |
| UCF [1] | loose | 38 | × | × |
| UCY [25] | loose | 4 | × | × |
| ETH [38] | loose | 2 | × | × |
| Subway Station [57] | loose | 1 | × | × |
| CUHK Crowd [44] | medium | 474 | 8 | × |
| Collective Motion [56] | medium | 413 | × | × |
| Marathon [2] | dense | 3 | × | × |
| DCFD (ours) | dense | 457 | 6 | ✓ |

### 4.2 Training Details

We implement our HINN with PyTorch [36] and conduct all the experiments on a single Nvidia GeForce RTX 3090 GPU. The number of simulation time steps $T$ in training and testing phase is 100. The relaxation time $\tau$ in Eq.10 is set to 1. The parameters $\lambda_1, \lambda_2, \lambda_3, \lambda_4, \lambda_5$ in $f_{acc_x}$ and $f_{acc_y}$ are 1, -0.2, 0.5, 1 and 1 respectively. Here, we select this group of parameters out of a simple comparison. Particularly, $\lambda_2$ is negative since the convection acceleration and the viscosity force for fluids lie on either side of Navier-Stokes equations as shown in Eq.2. The training set and the test set are split from DCFD at a ratio of seven to three. All experiments are conducted with ADAM [22] as the optimizer, Huber loss as the loss function, and IS, FID and SSIM as the evaluation metrics. More details about the parameters can be seen in the supplementary.

### 4.3 Experimental Results

To examine the effectiveness of our proposed model, we compare it with two kinds of simulation methods. One method is to treat crowds as assemblies of individuals and constrain their behaviors with rules, which can effectively reflect the social properties of crowds. For this kind of method, we select traditional models such as Boids and SFM based on their established performance in crowd simulation. The other method is to guide crowd motions with the original Navier-Stokes equations and solve the governing equations with various PDE-solving networks, which focuses on the physical properties of crowds. For this kind of method, we explore different types of PDE-solving networks, including PDE-Net (data-driven), NSFnets [21] (physics-informed), and the advanced PPNN. In order

[1]https://www.gettyimages.com

**Table 2: Comparative performance of simulation models evaluated by Inception Score (IS), Fréchet Inception Distance (FID), and Structural Similarity (SSIM).**

| Model | Boids | SFM | PDE-Net | NSFnets | PPNN | HINN (ours) |
|---|---|---|---|---|---|---|
| IS ↑ | 1.327 | 1.367 | 1.481 | 1.212 | **1.770** | 1.721 |
| FID ↓ | 0.111 | 0.144 | 0.195 | 0.709 | 0.064 | **0.056** |
| SSIM (%) ↓ | 41.27 | 45.91 | 58.76 | **32.14** | 38.10 | 37.27 |

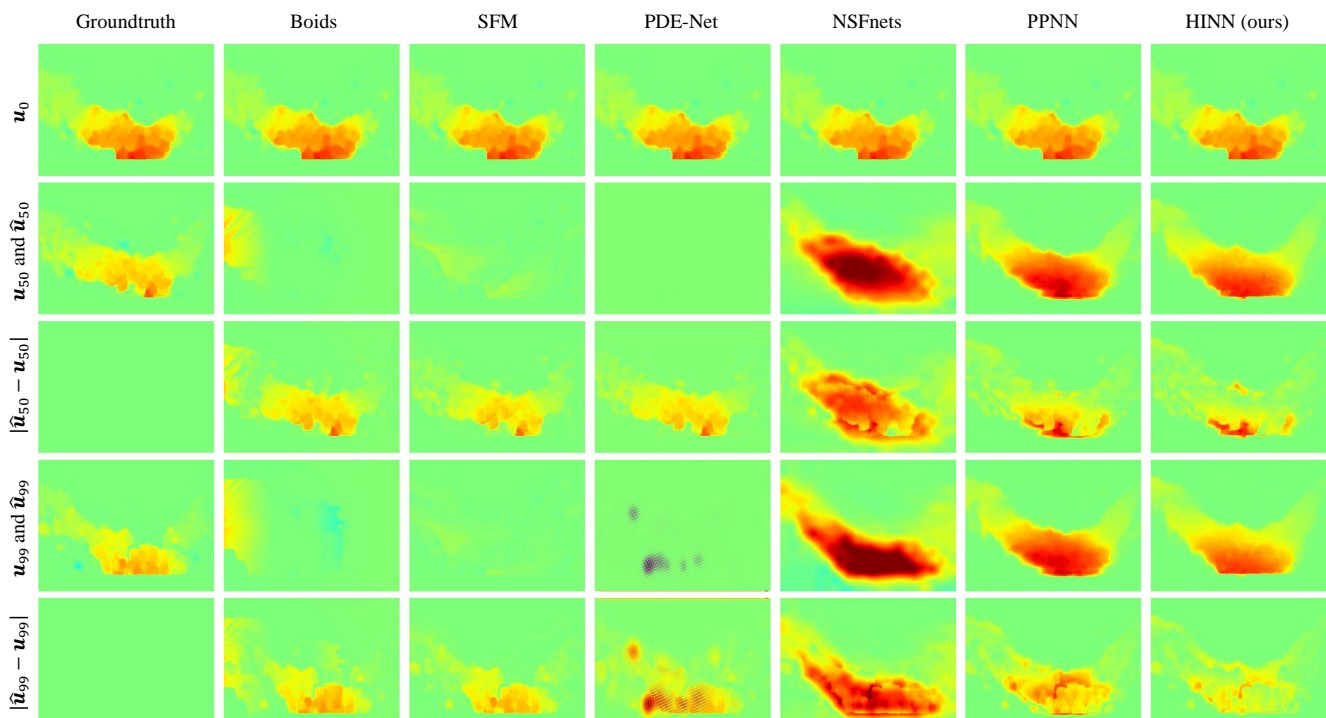

**Figure 3: Comparison between groundtruth and simulations of the horizontal component of crowd motion under a "curve" pattern at different time-steps. The first row depicts the given initial state. The second and fourth rows present the groundtruth and corresponding simulations at time-step 50 and time-step 99 respectively. The third and fifth rows display error maps for each model compared to the groundtruth at time-steps 50 and 99.**

to conduct the comparison, we apply our HINN along with these five simulation models to simulate dense crowd motions under six classic motion patterns and evaluate them with IS, FID and SSIM.

According to the data presented in Tab.2, our proposed HINN outperforms all the other simulation models in general, achieving remarkable results in terms of IS, FID and SSIM. Considering classic crowd simulation models, namely Boids and SFM, we can see that they exhibit mediocre performance across all evaluated metrics, which again demonstrates that microscopic models are not suitable for simulating dense crowds. PDE-Net suffers from mode collapse that it fails to generate crowd motions under "cross" and "scatter" patterns, which contributes to its poor performance on groundtruth-related metrics such as FID and SSIM. NSFnets performs slightly better than our HINN on SSIM, but shows disastrous performance on IS and FID, indicating its inadequacy in authenticity, fidelity and diversity. Closest to our HINN, PPNN shows better performance

on IS but falls behind on FID and SSIM, suggesting that while it effectively captures the crowd physical properties like continuity and fluidity, it struggles to maintain authenticity and fidelity in the simulation. Compared to these models, our HINN excels in terms of FID and are able to achieve comparable performance on IS and SSIM relative to the best-performing models.

In order to compare the performance of the simulation models more intuitively, we present the groundtruth and corresponding simulations using heat maps. As shown in Fig.3, the groundtruth is generated from a dense crowd video that contains crowd motion under a "curve" pattern. Given the initial state $\mathbf{u}_0$, Boids, SFM and PDE-Net can barely simulate the crowd motion after 50 time-steps. NSFnets merely restores the approximate overall shape, while overlooking finer details of dense crowd motions. The performance of PPNN at time-step 50 is actually acceptable. However, it appears that the simulation of PPNN remains relatively unchanged over

time, resulting in a larger error at time-step 99. Compared to these simulation models, our HINN successfully simulates the entire process of this "curve" motion, pays attention to intricate details of crowd motions, and demonstrates its adaptability over time. The simulations of more crowd motions under different motion patterns can be found in the supplementary.

## 4.4 Ablation Study

In our governing equation for dense crowd simulation, we consider both the physical properties and social properties of crowds. To investigate the contribution of these two kinds of properties in the dense crowd simulation, ablation studies are conducted for physical and social operators. (1) w/o physical operators. We erase the physical operators ($f_{con}$ and $f_{vis}$) from our governing equation, relying solely on the operators that highlight crowd characteristics distinct from fluids. (2) w/o social operators. We exclude the social operators ($f_{ali}$, $f_{nav}$ and $f_{coh}$) from our governing equation. In this circumstance, dense crowds are treated purely as fluids, neglecting the influence of crowd social behaviors. (3) w/o physical operators and social operators, i.e. w/o HIM. We remove the plug-in HIM which preserves our governing equation. In this scenario, our HINN reduces to a ConvResNet.

From data in Tab.3, it can be seen that after removing the physical and social operators from our governing equation, the simulation performance degrades across all evaluated metrics. Among all these models, HINN incorporated with physical operators demonstrates the highest similarity to our HINN in terms of SSIM, indicating the superior effectiveness of physical properties in maintaining structural integrity of crowd motions. HINN utilizing the social operators exhibits the closest alignment with our HINN based on FID and the average loss, underscoring the superiority of social properties in enhancing the fidelity and generating more realistic simulations. Additionally, either the physical operators or social operators may individually introduce a decrease in IS, but when we combine these two kinds of operators together in our governing equation, HINN achieves the best performance. To show influence of physical and social operators on dense crowd simulation more clearly, we exhibit a comparison of simulating crowd motion under a "scatter" pattern using different models as shown in Fig.4. We can observe that with physical operators (third and fifth rows), HINN better restores the overall motion and continuity of crowd; while with social operators (fourth and fifth rows), HINN can capture more finer details of crowd motions. Containing both physical and social operators, our HINN delivers the most comprehensive simulation with minimal average loss.

**Table 3: Contribution of the physical properties and social properties in dense crowd simulation measured by IS, FID, SSIM and the average loss (AL).**

| Model | IS ↑ | FID ↓ | SSIM (%) ↓ | AL ↓ |
|---|---|---|---|---|
| HINN w/o HIM | 1.704 | 0.060 | 38.12 | 8.653 |
| HINN w/o social operators | 1.698 | 0.065 | 37.32 | 8.543 |
| HINN w/o physical operators | 1.684 | 0.057 | 37.43 | 8.438 |
| HINN (ours) | **1.721** | **0.056** | 37.27 | **8.175** |

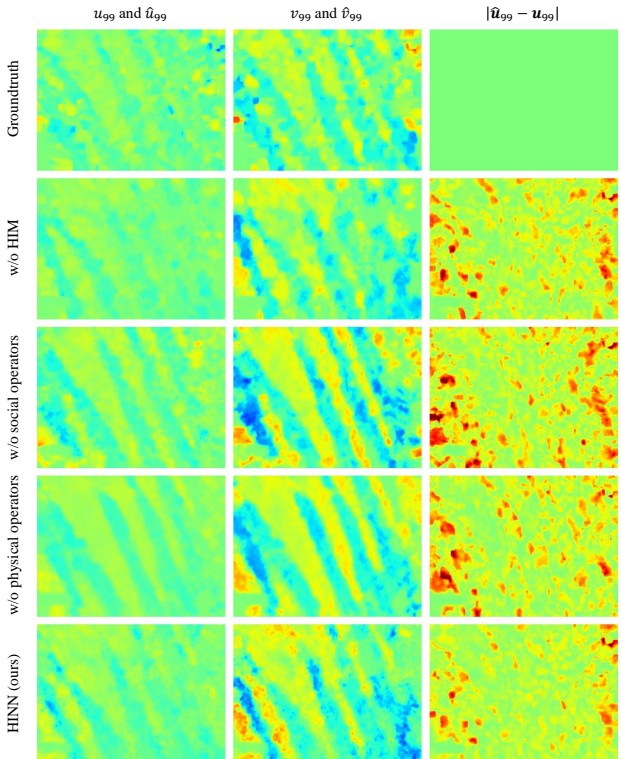

$u_{99}$ and $\hat{u}_{99}$  $v_{99}$ and $\hat{v}_{99}$  $|\hat{u}_{99} - u_{99}|$

Groundtruth / w/o HIM / w/o social operators / w/o physical operators / HINN (ours)

**Figure 4: Comparison between groundtruth and simulations of crowd motion under a "cross" pattern at time-step 99. The first and second column displays the horizontal and vertical components of the groundtruth and simulations respectively. The third column presents error maps of generated velocity for each model compared to the groundtruth.**

## 5 CONCLUSION

In this research, we propose a hydrodynamic model that addresses both the crowd physical properties and social properties for dense crowd simulation. Due to the challenge of directly solving the governing equation that is based on Navier-Stokes equations, we introduce the Hydrodynamics-Informed Neural Network (HINN) inspired by PPNN, which preserves the structure of our governing equation within the Hydrodynamics-Informed Module (HIM). To support the evaluation, we construct a new dense crowd motion video dataset called the Dense Crowd Flow Dataset (DCFD), which contains six classic crowd motion patterns. Based on DCFD, we utilize the objective metrics concerning authenticity, fidelity and diversity to evaluate performance of our model alongside other simulation models for simulating dense crowd motion patterns. Numerous experiments have demonstrated that our proposed model outperforms other simulation models and have shown the reliability of generated dense crowd motions. In future, we will continue to construct a more comprehensive dense crowd dataset with more video clips and motion patterns. We will also explore more physical and social operators which can reflect crowd properties so that we can simulate more complicated crowd motions.

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
