# OpenReview forum: "Hydrodynamics-Informed Neural Network for Simulating Dense Crowd Motion Patterns"
_acmmm.org/ACMMM/2024/Conference — MM2024 Poster_

### Official Review · Reviewer_tatN · 2024-05-21

**Rating:** 4
**Confidence:** 2

**Summary:**

Dense crowd simulation is crucial in addressing the increasing frequency of crowd crushes. In this research, the authors aim to simulate dense crowd movements under six motion patterns using an initial state. Drawing parallels with fluid dynamics, they propose a hydrodynamic model that captures both the physical and social properties of crowds through Navier-Stokes equations and crowd-interaction operators. To address computational challenges, they introduce the Hydrodynamics-Informed Neural Network (HINN), which solves the governing equations. Furthermore, the authors construct a new dataset, DCFD, to facilitate the evaluation of their approach.

**Strengths:**

1) Unique integration of hydrodynamics and neural networks for crowd simulation.
2) Solid mathematical basis using Navier-Stokes equations and social interaction operators.
3) The proposed HINN effectively solves the equations, addressing computational challenges.
4) New dataset, DCFD, enables comprehensive testing of the model.
5) Has the potential to improve crowd management, safety planning, and emergency response.

**Limitations:**

1) While a new dataset, DCFD, is introduced, the evaluation may be insufficient in terms of variety of scenarios, crowd densities, and comparison to state-of-the-art methods.
2) The paper may fail to adequately discuss potential limitations of the proposed approach, such as its performance under extreme conditions or generalization to real-world scenarios.
3) The choice of 6 patterns lacks justification compared to the 8 patterns in [44].
4) The rationale for using CNN as the baseline is unclear, especially given the advantages of Transformers.
5) References for comparison methods in Table 2 are missing, making it difficult to locate original literature.
6) The paper does not provide visualizations to illustrate the unique challenges and characteristics of the proposed new dataset. Without such visualizations, it is difficult for readers to fully understand the motivation for developing a new dataset and how it differs from existing ones.

**Suitability:**

3

---

### Official Review · Reviewer_wSrf · 2024-05-27

**Rating:** 3
**Confidence:** 3

**Summary:**

This paper proposes a new hydrodynamic model based on Naviser-Stokes equations and introduces a new dense crowd motion video dataset for crowd simulation. The hydrodynamic model can effectively capture physical and social properties by incorporating additional forces and achieve promising results on the new dataset.

**Strengths:**

1.	The authors contribute the largest dense crowd motion video dataset.
2.	The authors use more objective metrics to evaluate the performances of models.
3.	The structure of this paper is quite comprehensive.

**Limitations:**

1.	In the visualizations, when highlighting areas of improved performance in the color boxes, it would be helpful to understand.
2.	Do you have more detailed parameters of the dataset, in terms of the number of crowds, the diversity of the scenes, and so on?
3.	The paper does not explicitly discuss ethical considerations. While there may not be direct ethical implications, discussions on responsible data use and potential biases in crowd simulation algorithms would strengthen the paper.
4.	In Table 2, the proposed method performs slightly better than PPNN, but it is difficult to prove the effectiveness of operators that describe crowd interactions and crowd-environment interactions. Have you conducted more experiments on other existing datasets?
5.	Have you conducted simple cross-scene experiments on the proposed algorithms?

**Suitability:**

2

---

### Official Review · Reviewer_kUmq · 2024-05-28

**Rating:** 4
**Confidence:** 3

**Summary:**

This paper introduces a Hydrodynamics-Informed Neural Network (HINN) for simulating dense crowd motion patterns. It addresses challenges in dense crowd simulation: the nonlinear complexity of directly solving Navier-Stokes equations, the oversight of crowd characteristics distinct from fluids, and the lack of robust evaluation for crowd simulation models. To overcome these, it integrates physical and social properties in the Navier-Stokes equation and introduces corresponding operators into the PDE-Solving neural network HINN. The HINN preserves the governing equation structure in its architecture, enhancing computational efficiency. A new real-world dense crowd motion video dataset is created for evaluation. New objective metrics about authenticity, fidelity, and diversity are employed, including IS, FID, and SSIM. Experiments demonstrate the superior performance of HINN in simulating dense crowd motions compared to other models.

**Strengths:**

1. This paper clearly describes the traditional Navier-Stokes equations-based modeling of crowd dynamics and highlights its limitations and challenges, explaining the necessity of incorporating individual and social factors for crowd movement into the hydrodynamics equation.

2. The introduction of calculation elements for the velocity field in the governing equation is innovative. By introducing meticulously designed forces for *Alignment, Navigation, and Cohesion* on top of the existing *Convection and Viscosity* forces, the social properties of crowd movement are effectively represented.

3. This work contributes a new crowd motion video dataset, providing rich video data and velocity field statistics for various dense scenarios.

**Limitations:**

1. Subjectively, I find the methodological innovation of this paper to be somewhat lacking. The core HINN framework proposed in the text merely follows the PPNN scheme, with contributions focused on replacing the physical function with Navier-Stokes equations and introducing new social forces.

2. Equations 2, 3, and 4 in section 3.2 use pressure force, but it is not mentioned in subsequent parts. The paper should explain why this term from the Navier-Stokes equation was not applied.

3. There is a lack of discussion comparing the proposed method to deep learning crowd simulators. Although the paper compares with basic models like SFM, it should also discuss comparisons with state-of-the-art physics-informed deep learning models (e.g., [1][2]) to fully demonstrate the paper's opinion that microscopic models are not suitable for simulating dense crowds.

4. There are some textual errors in the paper: 1) In Figure 2, 'upsampling' should be 'bicubic'; 2) the definitions of N and M in equation 18 are not clearly provided.

[1] Social physics informed diffusion model for crowd simulation, AAAI24

[2] Human trajectory prediction via neural social physics, ECCV22

Also, I have remaining **questions** listed as below:
a. With a fixed equation form and a single given initial velocity, how does the predictive HINN generate diverse (i.e., different pattern) flows? What accounts for the good performance of IS?

b. What is the significance of evaluating IS? This metric is not related to the optimization objective in the problem formulation. It does not measure the distance between predicted and real samples. Shouldn't more general distribution metrics like MMD be considered?

c. Does the Canny edge detector operate on the velocity field or frames?

d. What does the 'average loss' in the ablation study refer to?

e. I do not find links to code and data. Would it be open-released?

**Suitability:**

3

---

### Meta-Review · Area_Chair_Uc6w · 2024-06-30

**Recommendation:** Accept (Poster)
**Confidence:** 5

**Metareview:**

This paper originally received 2 BA and BR, which became WA and 2 BA after rebuttal.

Main concerns mainly regarded confusing methodological parts to be clarified, weak novelty, missing discussion/analysis of similar comparative approaches, lack of discussion about ethical issues, insufficient experimental analysis and comparisons, insufficient analysis of the limitations of the proposed approach, and, more in general, other issues regarding weak or lack of comments or discussion and justifications of some methodological choices. Moreover, since a new dataset is also introduced, some remarks addressed the quality of this new data in terms of control parameters of such data.

Despite the number of criticisms, reviewers appreciate this work and the rebuttal satisfactorily addressed the majority of the raised remarks. For these reasons, this paper is considered acceptable for publication to ACM MM 2024.